# Research on Space Occupancy, Activity Rhythm and Sexual Segregation of White-Lipped Deer (*Cervus albirostris*) in Forest Habitats of Jiacha Gorge on Yarlung Zangbo River Basin Based on Infrared Camera Technology

**DOI:** 10.3390/biology12060815

**Published:** 2023-06-03

**Authors:** Yujia Liu, Kai Huang, Xueyu Wang, Ali Krzton, Wancai Xia, Dayong Li

**Affiliations:** 1Key Laboratory of Southwest China Wildlife Resources Conservation (Ministry of Education), China West Normal University, Nanchong 637009, China; liuyujia_2021@163.com (Y.L.); huangkai@ioz.ac.cn (K.H.); joexy460888599@163.com (X.W.); 2Key Laboratory of Conservation Biology of *Rhinopithecus roxellana* at China West Normal University of Sichuan Province, Nanchong 637001, China; 3Auburn University Libraries, Auburn University, Auburn, AL 36849, USA; alk0043@auburn.edu

**Keywords:** *Cervus albirostris*, infrared camera, activity rhythm, site occupancy model, sexual segregation

## Abstract

**Simple Summary:**

Infrared camera technology was used to study the space occupancy, activity rhythm, and sexual segregation of white-lipped deer (*Cervus albirostris*) in the Yarlung Zangbo River basin of Tibet for two years. The results showed that, in Jiacha Gorge, the occupancy of white-lipped deer is high, belonging to crepuscular activity pattern mammals. White-lipped deer have two activity peaks in the forest (May, October), and most of their activities are in the form of mixed groups of male and female during breeding and wintering. This study provides important information for the habitat utilization of white-lipped deer, and offers a scientific basis and theoretical support for the protection of white-lipped deer in Jiacha Gorge in the Yarlung Zangbo River basin.

**Abstract:**

The white-lipped deer (*Cervus albirostris)* is a rare and endangered species found in the Qinghai-Tibet Plateau in China. To understand the space occupancy, activity rhythm, and sexual segregation of the white-lipped deer, 24,096 effective photos and 827 effective videos were captured using infrared cameras from February 2020 to January 2022. The ecology and behavior of the white-lipped deer in Jiacha Gorge were studied in more detail using site occupancy models, relative abundance index, and other technologies and methods. The results show that The occupancy predicted by the model exceeds or approaches 0.5. The occupancy increases with greater altitude and with larger EVI values, while the detection rate increases with altitude only during spring and decreases with EVI values only in summer. The daily activity peaks for white-lipped deer were observed from 7:00 to 11:00 and 17:00 to 22:00, with annual activity peaks occurring from April to June and from September to November. From July to the following January, white-lipped deer mostly move in mixed-sex groups, while during the remainder of the year, they predominantly associate with individuals of the same sex. Climate, vegetation coverage, food resources, and human disturbance collectively influenced the behavior and habitat utilization of white-lipped deer. The foundational research conducted on white-lipped deer over the past two years is expected to enhance the basic understanding of white-lipped deer in the Qinghai-Tibet Plateau and contribute to future protection and management decisions.

## 1. Introduction

In the past 20 years, infrared camera technology has rapidly developed and become an effective means to study the ecology of various large and medium-sized mammals [1,2]. It is widely used in studies regarding species distribution [3], population density [4], relative abundance [5], activity rhythm [6], and spatial occupancy [7].

The occupancy model proposed by Mackenzie et al. [8] estimates the detection rate by establishing repeated detection to compensate for low detection probabilities. This model can estimate the spatial occupancy of the target species and their dynamics based on presence/absence data surveyed under “imperfect detection” conditions (i.e., detection probability < 1). In recent years, site occupancy models have been widely used in the analysis of biodiversity observation data [9,10], including infrared camera data, to indicate the spatial distribution of large and medium-sized mammals [11]. Studies using infrared camera data for occupancy detection mainly focus on species distribution [12], species and habitat relationships [13,14], and interspecies relationships [15].

Animal activity rhythm refers to the intensity of activity and changes in the activity patterns of animals at different times, seasons and places [16]. It reflects animals’ behavioral adaptations to the periodicity of environmental conditions (including biotic and abiotic factors) and their own physiological conditions [17]. Infrared camera technology has been widely used to research the activity rhythms of wild animals such as cougars (*Puma concolor*) [17], pygmy rabbits (*Brachylagus idahoensis*) [18], and Asian black bears (*Ursus thibetanus*) [19].

Sexual segregation results from conspecific individuals feeding, resting, and moving together in single-sex groups during the non-breeding season. This phenomenon is widespread in vertebrates [20], especially ungulates [21,22]. Sexual segregation has been observed in Pere David’s deer (*Elaphurus davidianus*), American bison (*Bison bison*), Przewalski’s gazelle (*Procapra przewalskii*)*,* red deer (*Cervus elaphus*), reindeer (*Rangifer tarandus*), African buffalo (*Syncerus caffer*), white-tailed deer (*Odocoileus virginianus*), and bighorn sheep (*Ovis canadensis*) [20]. The proximate and ultimate causes of sexual segregation could include differences between males and females in predation risk, social preferences, forage selection, scramble competition, and activity budgets [20,23].

The white-lipped deer (*Cervus albirostris*) is a rare and threatened species unique to the Qinghai-Tibet Plateau in China and is classified as vulnerable by the IUCN [24]. White-lipped deer inhabit alpine meadows, shrubs and forests between 3500 and 5000 m above sea level and primarily on grasses and sedges. Due to the remote geographical location and extremely harsh environment of their habitat, systematic field investigations regarding white-lipped deer (a large cervid) behavior are limited. Existing studies have evaluated the species’ taxonomy and distribution [25], feeding and community structure [26], and reproduction [27]. This study utilized 20 months of camera trapping data on white-lipped deer in order to: (1) model site occupancy for the species to evaluate the influence of environmental change; (2) survey natural activity patterns; (3) detect sexual segregation in *C. albirostris*.

## 2. Materials and Methods

### 2.1. Study Area

The study area is located in Jiacha Gorge, within the Yarlung Zangbo River basin of the Qinghai-Tibet Plateau. Jiacha Gorge is the second largest gorge carved by the Yarlung Zangbo River basin, accounting for 2.6% of the river’s total length [28]. The region has a large elevation drop (ranging from 3200 to 6000 m), with an average altitude of over 4000 m, average annual temperature of 8 °C, an extreme low temperature of −18.6 °C, and the annual average precipitation is 400 mm. The majority of precipitation occurs mainly from May to September due to the influence of the plateau temperate monsoon semi-humid climate [28,29,30]. The local vegetation is representative of the montane ecosystems of China, with vertical zonation as follows: mid-mountain evergreen/semi-evergreen broadleaf forest, subalpine evergreen coniferous forest, alpine shrub meadow and alti-frigetic subnival vegetation [31]. The vegetation type and vegetation cover of alpine shrub meadow may be affected by human factors such as grazing [32]. According to prior research and the existing literature, the Yarlung Zangbo River basin is located at the confluence of the Palaearctic and Oriental faunal regions, which contain many protected animals, including predators of white-lipped deer such as Leopards (*Panthera uncia*), Dholes (*Cuon alpinus*), Wolves (*Canis lupus*), Brown Bears (*Ursus arctos*), and competitors such as Alpine musk deer (*Moschus chrysogaster*), Chinese Serow (*Capricornis milneedwardsii),* and Himalayan serow (*Capricornis thar*) [33].

### 2.2. Infrared Camera Setup

The locations for the infrared cameras were determined based on factors such as uniformity of placement, altitude and other factors [34]. As altitudes above 5000 m in the study area are covered with snow year-round, to divide the area below 5000 m, was divided into 500 m × 500 m grid using ArcGIS grid. According to the grid accessibility, vegetation type, EVI, etc., each infrared camera was placed in a suitable location within the grid. The cameras were placed near trails climbing up from the valley bottom. Additionally, the minimum distance between all cameras was ≥ 500 m. Infrared cameras were installed in areas away from human activity, animal trails, water sources and other areas where animal activity is frequent and easily observed [35]. Although a total of 85 cameras were deployed (Figure 1), only 80 of them functioned as expected. The impact of Enhanced vegetation index (EVI, on the distribution of white-lipped deer, where a general green vegetation area is 0.2–0.8 [36]) on the distribution of white-lipped deer was considered. The placement of infrared cameras was arranged based on EVI gradient sampling in the study area (43.53% of the grids EVI < 0.2, 56.47% > 0.2). In total, 36 cameras (including 3 that did not work) were deployed in grids with EVI < 0.2, and 48 cameras (1 did not work) were deployed in grids with EVI > 0.2. The uniform sampling of altitude, slope and other factors could not be fully satisfied due to the consideration of EVI sampling and grid accessibility. The essential information for each infrared camera, including its ID number, altitude and the latitude and longitude of its placement was recorded at the time of deployment. The infrared cameras operated continuously for 24 h a day and when triggered, each camera would capture three consecutive photos and ten seconds of video. Every 3–5 months, we checked the cameras and collected data.

### 2.3. Site Occupancy Model

The field investigation phase of this study took place from February 2020 to January 2022. This survey time was divided into four seasons (spring, summer, autumn and winter), with each season undergoing single-season domain analysis. Within each season, the survey was repeated once every 10 days, totaling 18 times per season over two years. The presence or absence of white-lipped deer was recorded as binary data in the form of “0” and “1”. For example, the sequence “01001” would indicate that the target species was detected only on the second and fifth samples. The formula for calculating the likelihood result is as follows:Pr (01001) = *ψ* (1 − *p*_1_) *p*_2_ (1 − *p*_3_) (1 − *p*_4_) *p*_5_

If the detection result is “00000”, the formula for calculating the likelihood result is:Pr (00000) = *ψ* (1 − *p*_1_) (1 − *p*_2_) (1 − *p*_3_) (1 − *p*_4_) (1 − *p*_5_) + (1 − *ψ*)

In the formula, *ψ* represents occupancy and p represents the detection rate of the five repeated probes (*p*_1_, *p*_2_, *p*_3_, *p*_4_ and *p*_5_). Survey results were determined using on the maximum likelihood method.

Occupancy and detection are two crucial parameters in the site occupancy model. By default, these two parameters are assumed to remain constant when predicting site occupancy. However, in actual survey data, these two parameters are influenced by environmental variables such as altitude and the enhanced vegetation index (EVI) in the survey area. Therefore, it is necessary to incorporate these environmental variables into the model to ensure accurate evaluation. In this study, altitude, slope, intensity of human disturbance and (EVI) were used as the environmental variables affecting the coverage rate, while altitude and EVI were used as covariates affecting the detection rate. EVI data were derived from the MODIS data series regularly released by NASA. The intensity of human disturbance was calculated by determining the relative abundance index (RAI) of livestock and people observed during the survey period [37,38]. The formula is:*RAI* = *N*/*T* × 1000
where *N* stands for the number of independent effective probes, *T* stands for total camera workdays.

The MuMIN, permute and lattice packages in R were used to standardize the data to get more standard model results. The unmarked package was used to fit hierarchical models of animal occurrence and abundance to the imperfectly detected data. These packages were employed to analyze the occupancy model [35]. The best-fit models were selected using the Akaike Information Criterion (AIC) [39,40]. Based on the ranking of AIC values in the results, three models were identified as optimal. The optimal model had a ∆AIC less than 2.

### 2.4. Activity Rhythm Analysis

Bio-Photo V2.1 was used to pre-process the infrared camera data. The time-period relative abundance index (ITRA) was used to calculate the ITRA value of white-lipped deer each hour and month. The calculation formula is as follows:ITRA = Tij/Ni × 100
where Tij represents the number of effective photos of animals in class i (i = 1) during the time period (j = 0:00–0:59, 1:00–1:59… or January...December) and Ni represents the total number of individually valid photos of the i^th^ animal (i = 1) [41,42]. Based on the calculation results, circular histograms of daily activity rhythm and annual activity rhythm were constructed. Independent sample *t*-tests were performed to assess the difference in ITRA between the peak and off-peak periods of daily activity rhythm. A paired sample *t*-test was used to conducted the difference in ITRA of daily activities among different seasons.

### 2.5. Sexual Segregation Analysis

Clusters of individual deer were categorized as follows: male same-sex group, female same-sex group, and mixed-sex group. The mixed-sex group included adult mixed-sex group (consisting of adult male and adult female) and nursery mixed-sex group (comprising adult females and their offspring of different sexes). Throughout most of the year, white-lipped deer of the same sex (male or female) gather together to form the same-sex group. During the mating and wintering season, individuals of different sexes come together to form a larger mixed-sex group. After the breeding period, female white-lipped deer give birth to fawns and nurse them. Therefore, during this time, females with dependent fawns form mixed groups. Using photos and videos collected by the infrared camera, the herd were observed and classified based on group composition in different months. After calculating the frequency of male same-sex groups, female same-sex groups, and mixed-sex groups each month, Excel was used to visualize patterns of segregation between female and male deer when the white-lipped deer population is active.

## 3. Results

### 3.1. Site Occupancy Model

Between February 2020 and January 2022, the 80 functional infrared cameras deployed in the Jiacha Gorge captured photos and videos of the target species for 1490 days. A total of 49 cameras captured white-lipped deer, resulting in a grid occupancy of 0.613. The grid occupancy in summer, autumn, and winter is 0.438, 0.563, 0.525 respectively. When comparing the estimated occupancy by the model (Table 1) with the grid occupancy value, the estimated occupancy for each season closely aligns with the grid occupancy value. Table 1 displays the predicted detection rate according to the model. Based on the predictions for the environmental variables in the models of the occupancy and detection rate models, the occupancy of white-lipped deer across all four seasons increases with higher altitude and larger EVI values. However, the detection rate only increases with altitude in spring and decreases with elevation in the other three seasons. Additionally, the detection rate decreases with higher EVI values in summer but increases with EVI values in all other seasons (Table 2).

### 3.2. Activity Rhythm

Once the data from the infrared cameras were collected and sorted, a total of 24,096 usable photos and 827 usable videos of white-lipped deer were obtained. The analysis revealed two peaks in the daily activity rhythm of the white-lipped deer, occurring between 7:00-11:00 and 17:00-22:00. After the first peak of activity, deer significantly reduced their activity during the afternoon. The second peak of daily activity began in the early evening at around 17:00 and lasted until 22:00 (Figure 2). The ITRA (Mean ± SD = 7.634 × 10^−6^ ± 1.177 × 10^−5^) of the two peaks was significantly higher than the off-peak period (Mean ± SD = 3.695 × 10^−6^ ± 4.565 × 10^−6^; Independent *T* test, *F* = 7.718, *p* = 0.006). There were differences in the ITRA between each month, but not all month-to-month comparisons showed significant differences (refer to Table A1). The annual activity rhythm exhibited two peak periods (from April to June and from September to November) (Figure 3). Additionally, each season displayed two activity peaks in their daily activity rhythm (Figure 4). Although the peak time of daily activity of white-lipped deer varied across different seasons, the difference was not significant (refer to Table A2).

### 3.3. Sexual Segregation

The group activity of white-lipped deer can be classified as either same-sex grouping or mixed-sex grouping. The peak period for the observing mixed-sex groups occurred from July to January of the subsequent year, with January being the peak. From February to June, white-lipped deer are more likely to form groups with members of the same sex (Figure 5).

## 4. Discussion

Based on the site occupancy model, we found that the estimated occupancy rate closely aligns with the grid occupancy. The model’s predictions indicate that, in ten randomly selected samples, on average, more than five samples are expected to slow occupancy by white-lipped deer when the estimated occupancy rate exceeds or approaches 0.5. It is worth noting that human interference has become one of the influencing factors in the optimal model results during autumn. This may be because herdsmen successively transfer their grazing and animal husbandry activities during autumn, coinciding with the bearing period of many edible and utilizable plants for human. As a result, the frequency of human presence in the habitat of white-lipped deer significantly increases, leading to heightened human interference [43].

Among the three optimal models, altitude and Enhanced Vegetation Index (EVI) were identified as the main variables affecting model estimates. The occupancy rate of white-lipped deer increases with altitude. This could be due to several factors, such as the landscape at lower elevations consisting of dry-hot river valleys with sparse shrubs as the primary vegetation. Additionally, the presence of a provincial road along the river, may result in relatively low-quality and highly disturbed habitats at lower altitudes. Any combination of these factors may reduce the suitability of these areas for white-lipped deer. Furthermore, global climate change has caused warmer seasons in the Qinghai-Tibet Plateau, prompting white-lipped deer to migrate to higher altitude habitats due to adaptation [44].

In this survey, the infrared camera were only set below the forest line, which may have impacted the survey results. For instance, the observation that the detection rate increased with altitude only in spring could be attributed to winter snow cover at higher elevations causing white-lipped deer to avoid those areas. When spring returns and grasses and sedges (the preferred foods of white-lipped deer) become abundant in alpine meadows and alpine shrub habitats, the deer migrate back to previous altitudes [44].

The occupancy rate increased with higher EVI values regardless of season. Areas with high EVI exhibit high vegetation coverage, abundant plant growth, and preferred food sources for white-lipped deer. Within these habitats, white-lipped deer may feed on various plant species, including annual bluegrass (*Poa annua*), hammer sedge (*Carex hirta*), alpine bistort (*Polygonum viviparum*), Mongolian milkvetch (*Astragalus membranaceus*), and trees such as *Salix pseudotangii* and brown oak (*Quercus semicarpifolia*). Additionally, locations with high EVI may maintain intact vegetation coverage, serving as refuge areas for deer and other prey species, despite the presence of predators such as dholes (*Cuon alpinus*), wolves (*Canis lupus*), and snow leopards (*Panthera uncia*) in the area. However, the detection rate decreased with increasing EVI in summer but increased as expected in the other three seasons. This discrepancy may be due to the vegetation composition in the area, which mainly consists of evergreen coniferous forests (high EVI value). In early summer, shrubs and deciduous trees at lower altitudes begin to sprout, enticing white-lipped deer to feed in these areas (low EVI).

The daily activity rhythm of white-lipped deer is similar to that of other ungulates, meaning that peak activity generally occurs around sunrise and sunset [42,45]. The two peaks (7:00–11:00 and 17:00–22:00) in white-lipped deer resemble the daily activity pattern of other members of the Cervidae family, such as Tufted deer (*Elaphodus cephalophus*), Reeve’s muntjac (*Muntiacus reevesi*) [46], and Roe deer (*Capreolus capreolus*) [47]. One reason for this similarity may be that the grasses are covered with dew in the morning, providing water for the white-lipped deer. The weak light of dawn and dusk could also be conducive to avoiding predators [48]. Temperature is an important factor affecting the survival and distribution of organisms, which may affect internal body temperature, the intensity of metabolism, and the organism’s behavior [45]. White-lipped deer typically lie on their sides and chew cud or rest during the hottest part of the day, lowering their metabolic rate [43]. From 14:00 to 15:00 in the afternoon, the temperature is the highest throughout the day, and the high temperature during this period may be the main reason behind it being the least active period for white-lipped deer activity.

The annual activity rhythm of white-lipped deer also has two peaks, from April to June and again from September to November, which is similar to dwarf musk deer (*Moschus berezovskii*) [42]. These peaks may be related to the spatiotemporal distribution of food resources and the timing of reproduction, respectively. From April to June, the climate is relatively mild, and the broadleaf forest, shrub, coniferous forest and grasslands in the area provide abundant food for white-lipped deer. As the available feeding area increases, the intensity of deer activity also increases. Then, from September to November, it is the mating season for white-lipped deer. During this period, behaviors related to reproduction increase the activity of white-lipped deer. Examples include males joining groups of females and fighting for mating opportunities. Conversely, high temperatures in July and August force white-lipped deer to reduce their activity levels. Low temperatures and snow cover have a similar effect from December to March of the following year. Additionally, most herbs and low shrubs become buried under the snow, reducing food availability and forcing deer to conserve energy [45,49]. The regularity and variation of an animal’s activity rhythm is influenced by sympatric species such as humans, sympatric species, and abiotic variables such as climate and temperature in the area in which it lives [50]. The deviation of infrared camera distribution may be one of the reasons for the two annual activity peaks of white-lipped deer. This study only set infrared cameras below the forest line and failed to monitor white-lipped deer activities in alpine meadows and grasslands.

The activity rhythm of the same organism may change in different seasons [51]. The daily activity rhythm of white-lipped deer in different seasons has two peaks: the first peak is in the time range of 7:00–11:00, and the second peak is in the range of 17:00–22:00. The daily activity rhythm is one of the strategies for animals to adapt to the natural environment. The daily activity peak exists at different times in different seasons (7:00–8:00, 20:00–21:00 in spring; summer: 6:00–7:00, 21:00–22:00; autumn: 7:00–8:00, 19:00–20:00; winter: 9:00–10:00, 17:00–18:00) to accommodate seasonal changes in sunrise and sunset times [52]. The activity peak in the morning is relatively early in the summer and relatively late in winter. It is speculated that the reason for this may be that the sunrise time in summer is earlier, and the temperature rises quickly after the sun shines, so white-lipped deer choose to start foraging earlier. The sunrise in winter is later than in other seasons, and the morning temperature is low, reducing energy consumption in the low temperature environment. The activity peak is the second in the four seasons, occurring later in summer and earlier in winter. This is likely to be because daytime lasts longer in summer than in other seasons; thus, sunset is later. After sunset, the temperature drops quickly, and the dark of night helps white-lipped deer avoid predators. The activity peak is earlier in winter, which is presumed to be due to the earlier sunset compared to other seasons. After sunset, the temperature drops faster, leading to reduced white-lipped deer activity [53].

White-lipped deer are large social mammals that live in groups. While small groups typically comprise fewer than 20 individuals, these deer occasionally form large groups of more than 100 individuals. Male white-lipped deer have antlers with eight to nine branches, while females do not have any antlers. During the breeding period, when adult male deer begin to search for mates, the deer herd is composed of dozens of mixed-sex groups. However, female deer still form the majority within the herd, along with associated male deer up to 2-3 years old [48]. Outside the mating season, adult males and females tend to separate and move in single-sex groups, except for young males following their mothers [48]. The exception to this pattern is during the migration to the overwintering habitat, when female and male deer gradually intermingle to form a mixed-sex group [54].

There are significant differences in group activity types in different months. The data showed a higher frequency of mixed-sex groups in the study area from July to January of the following year, with peaks occurring in January and September. This finding is consistent with another study of white-lipped deer in a neighboring province, which showed that mixed-sex groups formed during the mating season from September to November [55]. Overwintering can also bring male and female deer back together more generally [54]. During the winter, when temperatures at the study site decreased, white-lipped deer in the Yarlung Zangbo River basin may overwinter in mixed groups similar to reindeer (*Rangifer tarandus*) and elk (*Cervus canadensis*) [27].

## 5. Conclusions

In this study, white-lipped deer were monitored using infrared cameras for two years. The results showed that the occupancy predicted by the model exceeded 1 or approached 0.5, while the overall detection rate was <0.5. The daily activity rhythm shows a bimodal trend, with a morning peak and an evening peak, and the annual activity rhythm also has two peaks. These findings for white-lipped deer are consistent with observations of some ungulates. The results also confirmed that white-lipped deer tend to form single-sex groups for most of the year except during the breeding season and while overwintering when mixed-sex groups predominate. Spatial occupancy of white-lipped deer is correlated with altitude, Enhanced vegetation index (EVI), human disturbance and other factors. This research provides a scientific basis and theoretical support for the protection of white-lipped deer in the Jiacha Gorge of the Yarlung Zangbo River basin.

## Figures and Tables

**Figure 1 biology-12-00815-f001:**
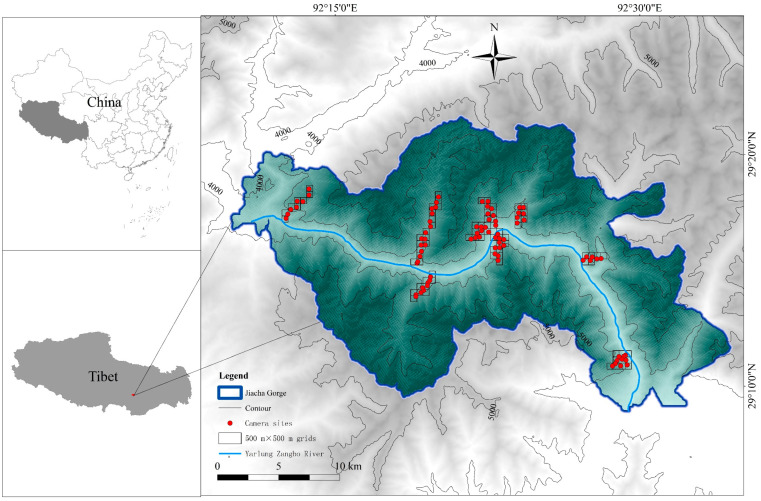
Locations of infrared camera traps in Jiacha Gorge.

**Figure 2 biology-12-00815-f002:**
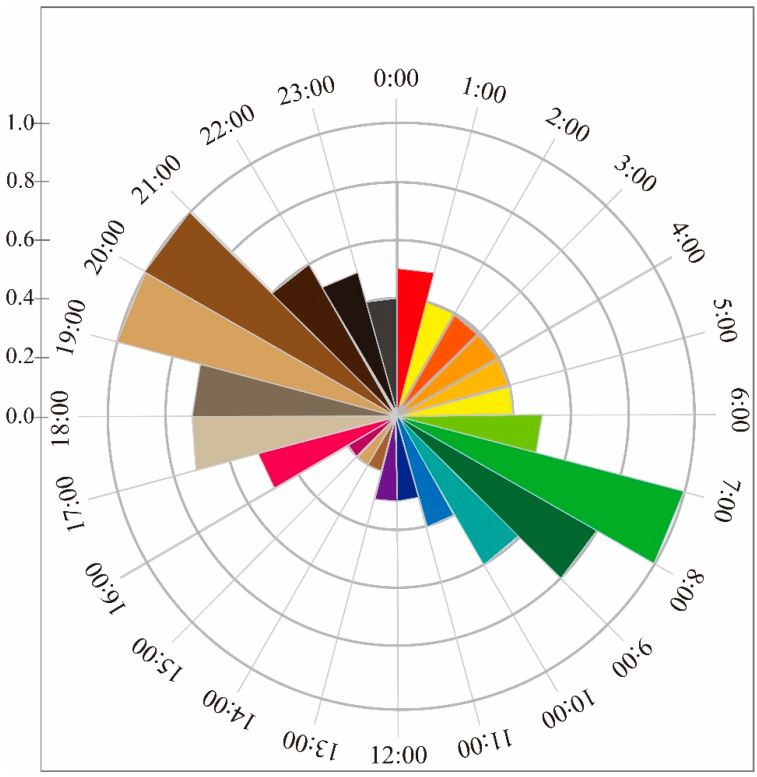
Daily activity rhythms of *Cervus albirostris.* The vertical coordinates are ITRA × 100,000.

**Figure 3 biology-12-00815-f003:**
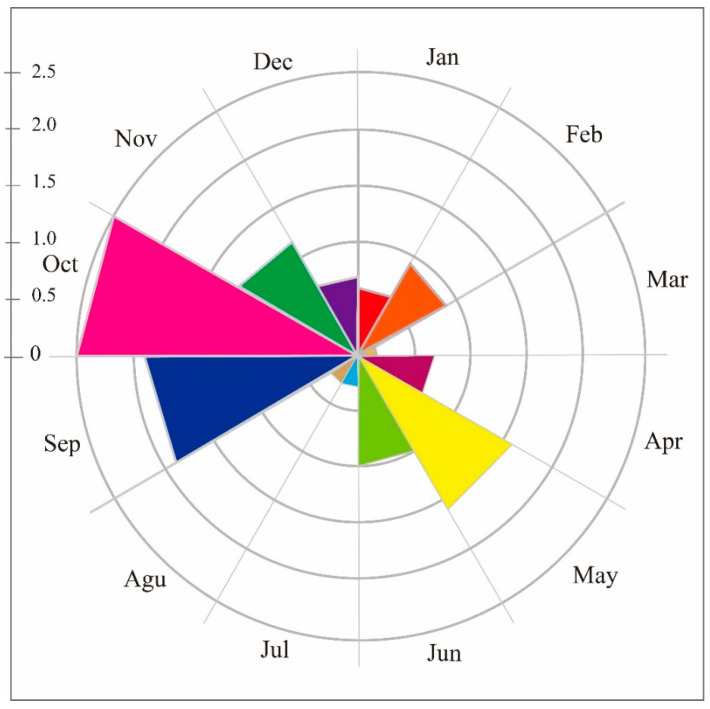
Annual activity rhythms of *Cervus albirostris.* The vertical coordinates are ITRA × 100,000.

**Figure 4 biology-12-00815-f004:**
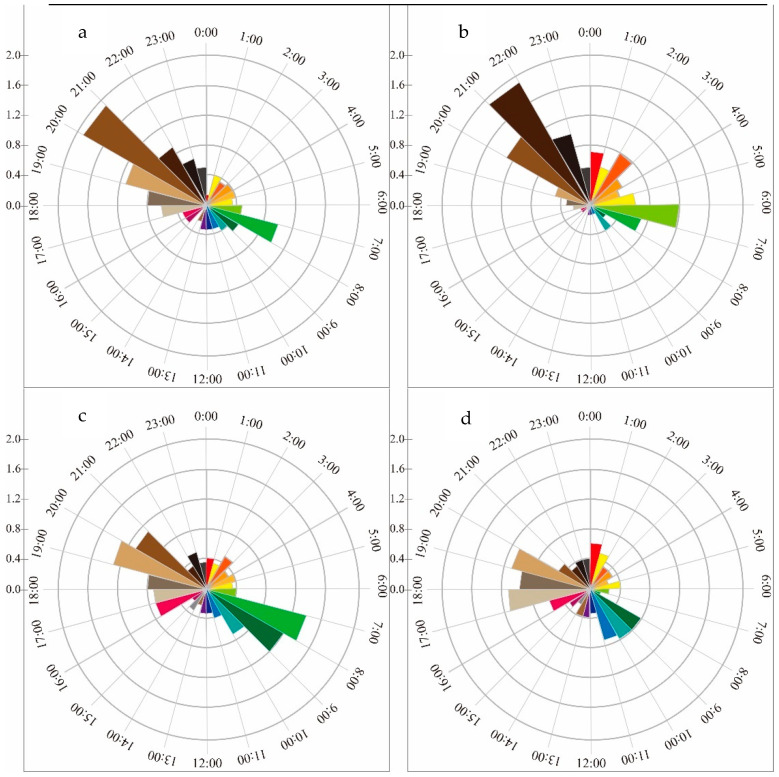
Daily activity rhythms of *Cervus albirostris* in different seasons. (**a**) spring, (**b**) summer, (**c**) autumn, (**d**) winter. The vertical coordinates are ITRA × 100,000.

**Figure 5 biology-12-00815-f005:**
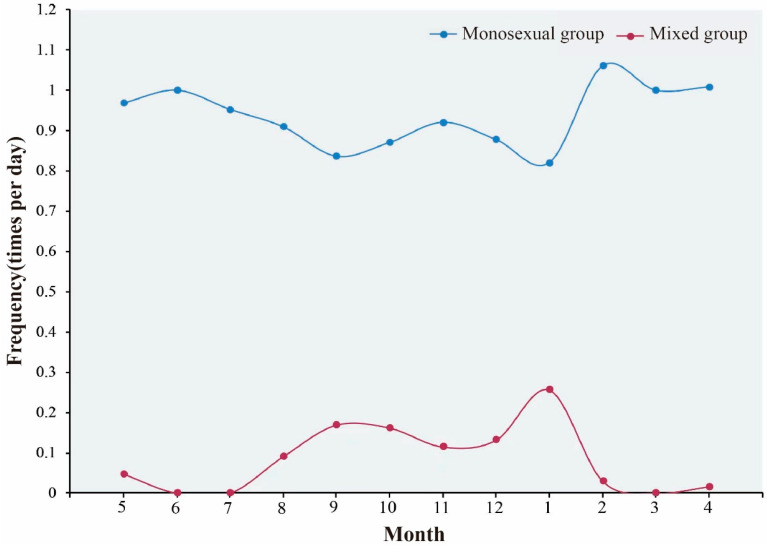
Rhythms of *Cervus albirostris* group types.

**Table 1 biology-12-00815-t001:** *Cervus albirostris* site occupancy models (psi = ***ψ*** is the site occupancy) in the Jiacha Gorge of Yarlung Zangbo River basin.

Seasons	Model	Npars	*AIC*	∆*AIC*	*AICwt*	*ψ*	95% Conf. Interval	*p*	95% Conf. Interval
Spring	*p* (elevation, EVI) *ψ* (elevation)	5	911.330	0.000	0.180	0.624	0.480–0.747	0.166	0.127–0.216
	*p* (EVI) *ψ* (elevation)	4	911.410	0.083	0.170	0.618	0.478–0.739	0.111	0.091–0.135
	*p* (elevation) *ψ* (elevation, EVI)	5	911.530	0.202	0.160	0.618	0.452–0.761	0.171	0.136–0.214
	Model average					0.620	0.471–0.749	0.149	0.118–0.188
Summer	*p* (elevation) *ψ* (elevation)	4	684.440	0.042	0.140	0.445	0.300–0.591	0.193	0.143–0.257
	*p* (elevation, EVI) *ψ* (elevation)	5	684.740	0.345	0.120	0.447	0.273–0.626	0.201	0.139–0.281
	*p* (elevation) *ψ* (elevation, EVI)	5	685.700	1.306	0.070	0.45	0.301–0.597	0.193	0.142–0.258
	Model average					0.447	0.290–0.605	0.196	0.141–0.266
Autumn	*p* (elevation) *ψ* (human_interference_intensity)	4	1051.410	0.000	0.190	0.551	0.429–0.685	0.356	0.306–0.408
	*p* (elevation) *ψ* (elevation, EVI)	5	1051.500	0.096	0.180	0.552	0.416–0.679	0.356	0.306–0.408
	*p* (elevation, EVI) *ψ* (human_interference_intensity)	5	1051.770	0.369	0.160	0.551	0.429–0.685	0.347	0.285–0.414
	Model average					0.551	0.425–0.683	0.353	0.300–0.410
Winter	*p* (elevation) *ψ* (elevation)	4	859.380	0.000	0.360	0.551	0.429–0.685	0.207	0.167–0.254
	*p* (elevation, EVI) *ψ* (elevation)	5	859.800	0.420	0.290	0.551	0.381–0.713	0.205	0.152–0.257
	*p* (elevation) *ψ* (elevation, EVI)	5	860.890	1.510	0.170	0.553	0.407–0.687	0.207	0.167–0.254
	Model average					0.551	0.407–0.695	0.206	0.162–0.255

Npars: number of model parameters; AIC: Akaike: Information Criteria; ∆AIC: AIC Relative difference between the best model and each other model in the set; AICwt: Weight: AIC value; 95% conf. interval: 95% confidence intervals; *p*: The detection rate predicted by the model; EVI: Enhanced vegetation index; Model average: the weighted average.

**Table 2 biology-12-00815-t002:** Covariates influencing *Cervus albirostris* occupancy and detectability according to *β*-coefficients and associated standard errors (*SE*).

Seasons	Model-Component	Covariates	Estimate	*SE*	*p*
Spring	Occupancy	Intercept	0.746	0.324	0.021
		Elevation	1.531	0.399	<0.001
		EVI	0.100	0.229	0.663
	Detection	Intercept	−1.609	0.115	<0.001
		Elevation	0.146	0.105	0.164
		EVI	0.089	0.109	0.412
Summer	Occupancy	Intercept	−0.262	0.271	0.333
		Elevation	1.114	0.353	<0.001
		EVI	0.049	0.158	0.758
	Detection	Intercept	−1.432	0.125	<0.001
		Elevation	−0.236	0.157	0.134
		EVI	−0.054	0.105	0.603
Autumn	Occupancy	Intercept	0.259	0.266	0.330
		Elevation	1.213	0.324	<0.001
		EVI	0.426	0.357	0.232
	Detection	Intercept	−0.631	0.085	<0.001
		Elevation	−0.479	0.096	<0.001
		EVI	0.012	0.046	0.786
Winter	Occupancy	Intercept	0.260	0.262	0.321
		Elevation	0.979	0.298	0.000
		EVI	0.035	0.134	0.795
	Detection	Intercept	−1.372	0.099	<0.001
		Elevation	−0.287	0.102	0.004
		EVI	0.046	0.092	0.615

## Data Availability

Not applicable.

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
