# Peer review of "Research on Space Occupancy, Activity Rhythm and Sexual Segregation of White-Lipped Deer (Cervus albirostris) in Forest Habitats of Jiacha Gorge on Yarlung Zangbo River Basin Based on Infrared Camera Technology"

_biology, 2023, doi:10.3390/biology12060815_

Round 1

Reviewer 1 Report

This is an interesting article with new information about a less-studied species of conservation interest, based on a substantial field effort that provided a large data set of observations.

Unfortunately, the article does not describe the methods or results in sufficient detail to make the reader confident of the conclusions.  There are also many small errors in the presentation and interpretation.  Some of these are addressed in the details below and can be addressed in revisions.   Several major issues need to be addressed in the Methods or Discussion sections in revisions.

Major issues:

1)     The placement of cameras is a key parameter that will determine what conclusions can be drawn from the dataset.  It appears that cameras were placed “haphazardly” along trails, not in the recommended stratified random array, with stations located within a selection of grid squares randomly chosen from an array of grid squares covering all possible habitats.  The manuscript should provide as much information as possible about how camera stations were chosen.  The reader can then decide for themselves how worrisome the potential for biases might be.  The potential biases can probably not be corrected at this point.  What can be done is to clearly describe exactly what was done and openly discuss the potential biases and that could have resulted from the non-random placement.  For example, it is problematic that cameras were not set above tree-line (lines 243-244), even though this species is known for its use of the high elevation habitats there.  This will certainly bias the results, as is already admitted in the Discussion.  That should also be made clear at the outset in the Methods section.  In fact, I recommend revising the title to include this information, e.g.  “…white-lipped deer (Cervus albirostris) in forest habitats of the Jiacha Gorge on Yarlung Zangbo River…”

2)     The discussion of “actual occupation” suggests that the authors do not have a clear understanding of how the occupancy model works.  The model takes into account imperfect detection, while records of occupation of grid squares is no more than presence-absence data, which confounds true absence with lack of detection.

3)     The differences across seasons are potentially interesting, but may be biased because of the absence of records from higher elevations.  The significance of the differences observed cannot be judged from the data presented in the paper. Seasonal values for Occupancy and Detection Probability should be presented together with their 95% Credible Intervals.  The plot shown, Figure 3, is not clearly explained, as the Methods section does not contain an explanation of how “density” might have been calculated.  Are these derived occupancy values or from trapping rates? Either way, it is likely that this reflects not activity but rather the migration of animals in and out of the area where cameras were set.  For example, as you state (lines 246-249), the deer range to higher elevations in the summer months, perhaps above tree line and out of the range of the camera array.  In mid-winter they may be moving lower, perhaps to other areas outside of the area set with camera traps.

4)     The impact of human disturbance is an extremely important issue.  The data suggests some important results about this, based on the significant influence of human_distrubance_intensity on Occupancy in the Autumn season.  This should be high-lighted and possible interpretations and implications discussed. 

5)     Elsewhere, the Discussion, conclusions are being made with little or no evidence presented to support them.  Basic data should be presented in a table, including the expected values of occupancy and probabilities of detection in each season together with their 95% credible intervals. This should be referred to, or additional statistical analysis should be presented, whenever a statement is made that there is a “significant impact” or “high correlation.”

6)     References:  Most of the references are older and not all of them are appropriate or up-to-date.  A literature search would lead the authors to many more current references and robust methods of analysis.

Detailed comments:

Lines 18-28, Simple Summary:  The first two sentences are generic and widely known, and they should be deleted from the Simple Summary.  Summarize the findings, not just the approach.  What was learned? What is important or new?

Line 49-50, first sentence of Introduction:  This is redundant with the 2nd sentence.  Combine the two and shorten.

Line 101-103, vegetation:  Provide a reference for the vegetation description, such as Peng 1997. 

Line 103:  Are alpine shrub meadow an anthropogenic-induced formation here, as they may be in Sichuan?   This possibility should at least be mentioned. See Miehe, G., Miehe, S., Bach, K., Nölling, J., Hanspach, J., Reudenbach, C., Kaiser, K., Wesche, K., Mosbrugger, V., Yang, Y.P., Ma, Y.M., 2011. Plant communities of central Tibetan pastures in the alpine steppe/Kobresia pygmaea ecotone. J. Arid Environ. 75, 711–723.

Line 108:  List important potential predators as well, or have they all been locally extirpated? (Apparently not, judging from mention below). Even if extirpated now, what would have been there in the past, potentially shaping the evolution of the deer?  Wolf, Wild Dog, Snow Leopard, Lynx, Brown Bear, Leopard?

Line 111:  “The intended locations for the infrared cameras were determined according to uniformity of placement, altitude and other factors.” What "other factors"?  This is critical information for reproducibility. Also, what was the distance between nearest cameras? 

Line 111: Were camera traps placed in a systematic array, randomly within grid squares, or haphazardly? The Methods section seems to imply that they were set based in an array based on a 500x500 meter grid, which would be logistically difficult in this steep terrain but appropriate.  However, the map in Figure 1 suggests that cameras were simply set at intervals along trails climbing up from the bottom of the gorge.  If it was not random placement, then the Discussion needs to include a short admission of the possible impacts of potential biases in sampling.  For example, were the same proportion of cameras placed in high EVI sites as low EVI sites?  If not, then the sample may be biased towards one or the other.  Based on a statement in the Discussion in Lines 243-244, it seems that few or no cameras were placed at the highest elevations, above tree-line. These biases probably cannot be fixed in retrospect, but at least they can be honestly described, and possible issues can be discussed here and in the Discussion (as done for elevation in lines 243-244).

Line 133:  Error in formula.  It should be the sum with (1-Psi), not the product.   Check the original in Xiao WH et al. 2019

Line 149:  Error in formula.  It should be "Number of independent detections (a.k.a. events)/ Total camera trap workdays x 1000"

Line 151:  Here and throughout, the word “occupation” should be replaced with the word “occupancy,” in the sense of MacKenzie, Nichols, Royle et al. 

Lines 150 – 153:  Provide more details here of the R packages used. In the interests of reproducible research, it would be better if you could provide your R scripts in Supplementary materials.

Line 165:  “Index of time-period relative abundance (ITRA)”

Line 178:  The meaning of mixed groups is unclear here.  Do you mean in mixed-sex groups or in groups of females with fawns mixed with females without fawns?

Line 191-192:  You write “actual occupation.”  How was this determined?  From presence or absence in blocks?  In that case, it does not take into account imperfect detection.  In keeping with convention, it should be called “naïve occupancy” to distinguish it from derived occupancy that is based on a model that does take into account detection probabilities less than 1.0.

Lines 192-193” The detection 192 rate predicted by the model is low;…”:  0.5 is not a low value for detection probability for 10 day session in a camera trap study. 

Lines 193-194:  The differences across seasons are potentially interesting, but are these differences significant?   These seasonal values, for Occupancy and Detection Probability, should be presented together with their 95% Credible Intervals. if not in the main text, then in Supplementary Materials.

Lines 197-198:  Since detection rate is influenced both by group density and by the probability of detection of an individual group, the impact of environmental variables on detection rate is tricky to decipher.  If both occupancy and detection increase with elevation, for example, it is likely that detection probability is increasing simply because group density increases with elevation, not because of changes in the probability of detection of an individual group.

Lines 206 to 208, Table 2:  You are presenting 4x2x3 = 24 P values here, so the risk of P-inflation is quite high.  You should use an algorithm like the Bonferroni correction or one of the less conservative alternatives to determine which P values are actually worth paying attention to. Those coefficients with P- values below the critical threshold are not worth the extensive discussion devoted to them below (e.g., all EVI coefficients). 

Line 214:  There were significant differences in the activity richness intensity of white-lipped deer in each month, with two peak periods…”  Please present the results of statistical tests here.  How is “activity richness intensity” measured?

Lines 217-218, Figure 2:  This data would be better presented as a circular plot. 

Lines 219-220, Figure 3:  How was this plot of annual activity rhythm generated?  From the occupancy analysis, or from raw presence-absence data?  Is it really “density” as stated, or should the y-axis be labeled “trapping rate”?  If it is from occupancy analysis, the axis should be labelled occupancy, and  it is likely that this reflects not activity but rather the migration of animals in and out of the area where cameras were set.  For example, as you state, the deer range to higher elevations in the summer months, perhaps above tree line.  In mid-winter they may be moving lower, or to another area outside of the area set with camera traps.

Line 231-233:  “The detection rate predicted by the model was less than 0.5; that is, in 10 randomly selected samples, less than 5 samples on average were 232 expected to show occupation by the white-lipped deer.”  What you define here is occupancy, not detection rate.  Detection rate is the probability of detection by the camera trap if the animals are actually present in the vicinity.

Lines 235-237:  “The occupation 234 rate of white-lipped deer was positively correlated with altitude in all seasons. The main reason for this is…Interesting conjectures follow, but these conclusions are not directly supported by the evidence that you have presented above.  Have you done any additional analysis to prove that these are the “main reason?”  Do you have additional evidence from your data or from the literature to support your conclusions?  If not, better to phrase it as “A possible reasons for this is….”

Lines 243-244:  “Placing cameras only in the forest and leaving areas above the treeline unsurveyed may have influenced the survey results…”  This is extremely important information, `which should also be presented in the Methods section.  This means that your study is only representative of the behavior of this species below tree-line.  In fact, it should be included in the title, which could be modified to included “… in subalpine forest habitats.” You should also address any other potential biases from the non-random camera trap station locations in the Discussion section.

Lines 258-260:  The detection rate conflates the probability of detection of an individual group and the density of groups.  As a result, it is tricky to interpret what changes in detection rate mean.  I do not think you would know a priori what to expect without some more information about the size and density of groups.  For example, in summer, at high EVI camera trap stations, the mid-story vegetation may be more dense, thus hindering detection of herds at a distance. Or the number of groups may be lower, perhaps because the deer are aggregating into a few large herds. At any rate, none of the coefficients for effect of EVI on detection are significant (Table 2), so it is hard to draw any conclusions at all.  It would be best to strike this discussion of detection rate entirely.

Line 263:  Insert some discussion here about the apparent impact of human_distrubance_intensity on Occupancy, but only in the Autumn season.  This might be one of the most important results from the study.  What is your interpretation?  Sept to November is the mating season.  What could be going on? 

Lines 273-274:  This is pure conjecture, so it should begin with “perhaps consequently….”

Line 276:  Seasonal “activity” was apparently measured based on trapping rates.  But this measures not only the movement of groups, but also the density of groups.  Fewer groups will mean a lower rate of detections.  If so, how can you separate activity from the conflated variable of group density, which will change as average group size changes?  And if deer are migrating out of the area sampled by your camera trap array, this will be recorded as lower “activity.”  For example, deer may be migrating to higher elevations above tree-line in summer, and this may be the cause of the decrease in “activity” that you claim in July and August.  

Lines 307-309:  “This study area, in the middle reaches of the Yarlung Zangbo River, has an average 307 altitude of over 4000 m, an average annual temperature of 8℃ and an extreme low temperature of -18.6℃, with heavy snow in winter.” This important information should be moved to the Methods Section, 2.1 Study Area.

Line 317:  “… and the annual activity rhythm is also bimodal.”   I am not convinced (see concerns above about Line 276.) Delete this phrase.

Lines 320-321:   “Spatial occupation of white-lipped deer is highly correlated with altitude, vegetation type, food availability, human disturbance, temperature, precipitation and other factors.”  In fact, your data only show that spatial occupancy is highly correlated with elevation, and also influenced by vegetation type and, seasonally, by human disturbance.  You have not presented any data or analysis to support any conclusions about influence of food availability, temperature, precipitation or other factors on occupancy or detection.

Line 325:  Please name your field assistants if possible.

References:  Most of the references are older and not all of them are appropriate or up-to-date. There are quite a few newer articles measuring activity patterns using camera trap data which would be worth checking, and perhaps cloning their methods to test for significance in patterns. E.g. check references and citations of Sandra Frey, Jason T. Fisher, A. Cole Burton, John P. Volpe 2017 in Remote Sensing in Ecology and Conservation (Open Access) Investigating animal activity patterns and temporal niche partitioning using camera-trap data: challenges and opportunities. https://doi.org/10.1002/rse2.60

There are numerous small mistakes and a general lack of clarity in the description of methods.  While these are grammatically correct, they suggest that the authors or translator did not fully understand how occupancy models and kernel analysis works.  For example, “actual occupation” appears to be used to describe naïve occupancy. The descriptions of methods would best be revised to mirror standard descriptions in the literature.

Author Response

Reply to the reviewer 1

It is a great honor for you to review my article so carefully and thank you for your valuable suggestions and professional questions. I have made corresponding modifications to your suggestions, and my reply to your questions is as follows:

Comment 1: The placement of cameras is a key parameter that will determine what conclusions can be drawn from the dataset.  It appears that cameras were placed “haphazardly” along trails, not in the recommended stratified random array, with stations located within a selection of grid squares randomly chosen from an array of grid squares covering all possible habitats.  The manuscript should provide as much information as possible about how camera stations were chosen.  The reader can then decide for themselves how worrisome the potential for biases might be.  The potential biases can probably not be corrected at this point.  What can be done is to clearly describe exactly what was done and openly discuss the potential biases and that could have resulted from the non-random placement.  For example, it is problematic that cameras were not set above tree-line (lines 243-244), even though this species is known for its use of the high elevation habitats there.  This will certainly bias the results, as is already admitted in the Discussion.  That should also be made clear at the outset in the Methods section.  In fact, I recommend revising the title to include this information, e.g.  “…white-lipped deer (Cervus albirostris) in forest habitats of the Jiacha Gorge on Yarlung Zangbo River…”

Response 1: Considering that the camera were not set above tree-line, it is may lead to the deviation of the results of activity rhythm. I accept your suggestion to change the title to: ”Research on space occupation, activity rhythm and sexual seg-regation of white-lipped deer (Cervus albirostris) in forest habitats of Jiacha Gorge on Yarlung Zangbo River basin based on infra-red camera technology”. And I also added instructions in the methods section and discussion section.

Comment 2: The discussion of “actual occupation” suggests that the authors do not have a clear understanding of how the occupancy model works.  The model takes into account imperfect detection, while records of occupation of grid squares is no more than presence-absence data, which confounds true absence with lack of detection.

Response 2: The actual occupation here should be the grid occupation (actual sampling grid occupancy in white-lipped deer). Such as in the spring there were 49 cameras captured white-lipped deer the grid occupation was 0.613. This research model is the domain occupation model developed by MacKenzie et al. (2002) based on the maximum likelihood method. The basic idea is to record whether the target species is detected through multiple surveys of unit samples in the study area. Undetected cases are explicitly dealt with and unbiased estimates of occupation.

Comment 3: The differences across seasons are potentially interesting, but may be biased because of the absence of records from higher elevations.  The significance of the differences observed cannot be judged from the data presented in the paper. Seasonal values for Occupancy and Detection Probability should be presented together with their 95% Credible Intervals.  The plot shown, Figure 3, is not clearly explained, as the Methods section does not contain an explanation of how “density” might have been calculated.  Are these derived occupancy values or from trapping rates? Either way, it is likely that this reflects not activity but rather the migration of animals in and out of the area where cameras were set.  For example, as you state (lines 246-249), the deer range to higher elevations in the summer months, perhaps above tree line and out of the range of the camera array.  In mid-winter they may be moving lower, perhaps to other areas outside of the area set with camera traps.

Response 3: 95% confidence intervals have been added to the table1. The results of daily and annual activity rhythms by time-period relative abundance index were consistent with those of nuclear density. So I modified the method part and draw a circular histogram of the activity rhythm using the calculated ITRA values. Additional variance analysis results are shown in the Appendix1 and Appendix2.

Comment 4: The impact of human disturbance is an extremely important issue.  The data suggests some important results about this, based on the significant influence of human_distrubance_intensity on Occupancy in the Autumn season.  This should be high-lighted and possible interpretations and implications discussed.

Response 4: Thanks for your professional advice, I have added the discussion at the end of the first paragraph of the discussion section. The influence of human disturbance on the occupation and detection rate in autumn may be due to the gradual change of herders' grazing in autumn. Autumn is also the growth period of many edible and available plants for human beings, which increases the frequency of human activities.

Comment 5: Elsewhere, the Discussion, conclusions are being made with little or no evidence presented to support them.  Basic data should be presented in a table, including the expected values of occupancy and probabilities of detection in each season together with their 95% credible intervals. This should be referred to, or additional statistical analysis should be presented, whenever a statement is made that there is a “significant impact” or “high correlation.”

Response 5: 95% confidence intervals of occupation and detection rate have been added to the table1. thank you for the reminder and I analyzed again the parts that put forward "significant impact" and "high correlation." and modified and adjusted them. I put in the statistical analysis I needed in the Appendix1 and Appendix2.

Comment 6: References:  Most of the references are older and not all of them are appropriate or up-to-date.  A literature search would lead the authors to many more current references and robust methods of analysis.

Response 6: Since there are few papers related to white-lipped deer, many articles on feeding habits, reproduction, and habitat selection of white-lipped deer were published in earlier years, so many references are older.

Detailed comments:

Comment: Lines 18-28, Simple Summary:  The first two sentences are generic and widely known, and they should be deleted from the Simple Summary.  Summarize the findings, not just the approach.  What was learned? What is important or new?

Response: Line 18-25, Simple Summary:According to your suggestion, I have modified and improved the Simple Summary.

Comment: Line 49-50, first sentence of Introduction:  This is redundant with the 2nd sentence.  Combine the two and shorten.

Response: Line 47-48, The two sentences have been fused into:”In the past 20 years, infrared camera technology has developed rapidly and be-come an effective means to study the ecology of various large and medium-sized mammals”.

Comment: Line 101-103, vegetation:  Provide a reference for the vegetation description, such as Peng 1997.

Response: I am very sorry that I did not find the corresponding literature according to the information you provided.

Comment: Line 103:Are alpine shrub meadow an anthropogenic-induced formation here, as they may be in Sichuan?   This possibility should at least be mentioned. See Miehe, G., Miehe, S., Bach, K., Nölling, J., Hanspach, J., Reudenbach, C., Kaiser, K., Wesche, K., Mosbrugger, V., Yang, Y.P., Ma, Y.M., 2011. Plant communities of central Tibetan pastures in the alpine steppe/Kobresia pygmaea ecotone. J. Arid Environ. 75, 711–723.

Response: Line 98-100: Most of the research area was naturally occurring landscapes. But the vegetation type and vegetation cover of alpine shrub meadow may be affected by human factors such as grazing. I mentioned this possibility in my article.

Comment: Line 108: List important potential predators as well, or have they all been locally extirpated? (Apparently not, judging from mention below). Even if extirpated now, what would have been there in the past, potentially shaping the evolution of the deer?  Wolf, Wild Dog, Snow Leopard, Lynx, Brown Bear, Leopard?

Response: Line 99-104: Take your advice and list the competitors and predators of white-lipped deer in the survey area.

Comment: Line 111:“The intended locations for the infrared cameras were determined according to uniformity of placement, altitude and other factors.” What "other factors"?  This is critical information for reproducibility. Also, what was the distance between nearest cameras?

Were camera traps placed in a systematic array, randomly within grid squares, or haphazardly? The Methods section seems to imply that they were set based in an array based on a 500x500 meter grid, which would be logistically difficult in this steep terrain but appropriate.  However, the map in Figure 1 suggests that cameras were simply set at intervals along trails climbing up from the bottom of the gorge.  If it was not random placement, then the Discussion needs to include a short admission of the possible impacts of potential biases in sampling.  For example, were the same proportion of cameras placed in high EVI sites as low EVI sites?  If not, then the sample may be biased towards one or the other.  Based on a statement in the Discussion in Lines 243-244, it seems that few or no cameras were placed at the highest elevations, above tree-line. These biases probably cannot be fixed in retrospect, but at least they can be honestly described, and possible issues can be discussed here and in the Discussion (as done for elevation in lines 243-244).

Response: Line 113-120 Thank you for your question. The placement of the camera is not random. In consideration of the effect of Enhanced vegetation index (EVI, general green vege-tation area is 0.2-0.8 [34]) on the distribution of white-lipped. The infrared camera was arranged according to EVI gradient sampling in the study area (43.53% of the grids EVI < 0.2; 56.47% > 0.2), 36 (3 not worked) cameras were deployed in grids with EVI < 0.2, and 48 (1 not worked) cameras were deployed in grids with EVI > 0.2. After consid-ering the sampling of EVI and the accessibility of grid, the uniform sampling of altitude, slope and other factors cannot be satisfied.

Comment: Line 133:  Error in formula.  It should be the sum with (1-Psi), not the product.   Check the original in Xiao WH et al. 2019

Line 149:  Error in formula.  It should be "Number of independent detections (a.k.a. events)/ Total camera trap workdays x 1000"

Line 151:  Here and throughout, the word “occupation” should be replaced with the word “occupancy,” in the sense of MacKenzie, Nichols, Royle et al.

Response: Line 139, 153-154: Thank you very much for your reminding, and I have corrected the error.

Comment: Lines 150 – 153: Provide more details here of the R packages used. In the interests of reproducible research, it would be better if you could provide your R scripts in Supplementary materials.

Response: Line 153-160: The specific purpose of the R package has been briefly explained in the article. R script comes from: Xiao, W. H.; Su, Z. F.; Chen, L. J.; Yao,W. T.; Ma,Y.; Zhang,Y. M.; Xiao, Z. S.; Using occupancy models in wildlife camera-trapping monitoring and the study case. Biodivers. Sci. 2019, 27, 249-256.

Comment: Line 165:  “Index of time-period relative abundance (ITRA)”

Response: Thank you for your correction, my article has corrected the error here.

Comment: Line 178: The meaning of mixed groups is unclear here.  Do you mean in mixed-sex groups or in groups of females with fawns mixed with females without fawns?

Response: Line 180-183: Mixed-sex group include adult mixed-sex group (adult male and adult female) and nursery mixed-sex group (consisting of adult females and their difference sex offspring).

Comment: Line 191-192:  You write “actual occupation.”  How was this determined?  From presence or absence in blocks?  In that case, it does not take into account imperfect detection.  In keeping with convention, it should be called “naïve occupancy” to distinguish it from derived occupancy that is based on a model that does take into account detection probabilities less than 1.0.

Response: The actual occupation here should be the grid occupation (actual sampling grid occupancy in white-lipped deer). Such as 49 cameras captured white-lipped deer in the spring the grid occupation was 0.163. The actual occupation here should be the grid occupation.

Comment: Lines 192-193” The detection 192 rate predicted by the model is low;…”:  0.5 is not a low value for detection probability for 10 day session in a camera trap study.

Response: Lines 196-204: Thanks for your advice, I revised the statement of low detection rate.

Comment: Lines 193-194: The differences across seasons are potentially interesting, but are these differences significant?   These seasonal values, for Occupancy and Detection Probability, should be presented together with their 95% Credible Intervals. if not in the main text, then in Supplementary Materials.

Response: Thanks for your advice, 95% confidence intervals of occupation and detection rate have been added to the table1.

Comment: Lines 197-198:  Since detection rate is influenced both by group density and by the probability of detection of an individual group, the impact of environmental variables on detection rate is tricky to decipher.  If both occupancy and detection increase with elevation, for example, it is likely that detection probability is increasing simply because group density increases with elevation, not because of changes in the probability of detection of an individual group.

 Lines 206 to 208, Table 2:  You are presenting 4x2x3 = 24 P values here, so the risk of P-inflation is quite high.  You should use an algorithm like the Bonferroni correction or one of the less conservative alternatives to determine which P values are actually worth paying attention to. Those coefficients with P- values below the critical threshold are not worth the extensive discussion devoted to them below (e.g., all EVI coefficients).

Response: The problem you mentioned is very scientific and professional, but it cannot be solved temporarily due to my lack of skills. I'll keep thinking about solutions.

Comment: Line 214: “There were significant differences in the activity richness intensity of white-lipped deer in each month, with two peak periods…”  Please present the results of statistical tests here.  How is “activity richness intensity” measured?

Response: Line 165-175 There were differences of the ITRA on each month, but not all month-to-month comparisons were significantly different. Measure activity richness intensity the number of independent valid photos and the total number of all independent valid photos of white-lipped deer in the corresponding period were sorted out and counted, and each ITRA value was calculated using the formula ITRA =Tij/Ni×100.

Comment: Lines 217-218, Figure 2:  This data would be better presented as a circular plot.

Lines 219-220, Figure 3:  How was this plot of annual activity rhythm generated?  From the occupancy analysis, or from raw presence-absence data?  Is it really “density” as stated, or should the y-axis be labeled “trapping rate”?  If it is from occupancy analysis, the axis should be labelled occupancy, and it is likely that this reflects not activity but rather the migration of animals in and out of the area where cameras were set.  For example, as you state, the deer range to higher elevations in the summer months, perhaps above tree line.  In mid-winter they may be moving lower, or to another area outside of the area set with camera traps.

Response: Thanks for your suggestion, the activity rhythm has been changed to a circular histogram display. Placing the camera below the forest line does present the problem you mentioned, I have added the explanation (line 313-336). However, since the experiment has been completed, this problem cannot be solved. I hope you can understand.

Comment: Line 231-233: “The detection rate predicted by the model was less than 0.5; that is, in 10 randomly selected samples, less than 5 samples on average were 232 expected to show occupation by the white-lipped deer.”  What you define here is occupancy, not detection rate.  Detection rate is the probability of detection by the camera trap if the animals are actually present in the vicinity.

Response: Line247-256: Thank you for finding my mistake, I have corrected it.

Comment: Lines 235-237:“The occupation 234 rate of white-lipped deer was positively correlated with altitude in all seasons. The main reason for this is…” Interesting conjectures follow, but these conclusions are not directly supported by the evidence that you have presented above.  Have you done any additional analysis to prove that these are the “main reason?”  Do you have additional evidence from your data or from the literature to support your conclusions?  If not, better to phrase it as “A possible reasons for this is….”

Response: I’m sorry that the expression positive correlation is wrong and I'm not doing correlation analysis here. However, it can be seen from Table 2 that the occupation of white-lipped deer increased with the elevation in all seasons. References: Xiao, W. H.; Su, Z. F.; Chen, L. J.; Yao,W. T.; Ma,Y.; Zhang,Y. M.; Xiao, Z. S.; Using occupancy models in wildlife camera-trapping monitoring and the study case. Biodivers. Sci. 2019, 27, 249-256.

Comment: Lines 243-244:  “Placing cameras only in the forest and leaving areas above the treeline unsurveyed may have influenced the survey results…”  This is extremely important information, `which should also be presented in the Methods section.  This means that your study is only representative of the behavior of this species below tree-line.  In fact, it should be included in the title, which could be modified to included “… in subalpine forest habitats.” You should also address any other potential biases from the non-random camera trap station locations in the Discussion section.

Response: Thank you for your suggestion. In my study I considering the sampling of EVI and the accessibility of grid, the uniform sampling of altitude, slope and other factors cannot be satisfied at the same time. I have explained the potential bias in camera placement.

Comment: Lines 258-260:  The detection rate conflates the probability of detection of an individual group and the density of groups.  As a result, it is tricky to interpret what changes in detection rate mean.  I do not think you would know a priori what to expect without some more information about the size and density of groups.  For example, in summer, at high EVI camera trap stations, the mid-story vegetation may be more dense, thus hindering detection of herds at a distance. Or the number of groups may be lower, perhaps because the deer are aggregating into a few large herds. At any rate, none of the coefficients for effect of EVI on detection are significant (Table 2), so it is hard to draw any conclusions at all.  It would be best to strike this discussion of detection rate entirely.

Response: Your advice is very professional and scientific. This is a problem that I have ignored but I’m sorry I cannot solve the problem you mentioned at present.

Comment: Line 263:  Insert some discussion here about the apparent impact of human_distrubance_intensity on Occupancy, but only in the Autumn season.  This might be one of the most important results from the study.  What is your interpretation?  Sept to November is the mating season.  What could be going on?

Response: Your suggestion is very timely. I did not discuss about it. So I added related discussion in Line 247-256.

Comment: This is pure conjecture, so it should begin with “perhaps consequently….”

Response: I took your advice and changed it to: “perhaps consequently….”

Comment: Line 276:  Seasonal “activity” was apparently measured based on trapping rates.  But this measures not only the movement of groups, but also the density of groups.  Fewer groups will mean a lower rate of detections.  If so, how can you separate activity from the conflated variable of group density, which will change as average group size changes?  And if deer are migrating out of the area sampled by your camera trap array, this will be recorded as lower “activity.”  For example, deer may be migrating to higher elevations above tree-line in summer, and this may be the cause of the decrease in “activity” that you claim in July and August. 

Response: You have given professional advice, but I have not solved this problem for the time being. The effect of camera placement on the results has been explained and its shortcomings have been acknowledged in my paper. Line 157-160, 313-318.

Comment: Lines 307-309:  “This study area, in the middle reaches of the Yarlung Zangbo River, has an average 307 altitude of over 4000 m, an average annual temperature of 8℃ and an extreme low temperature of -18.6℃, with heavy snow in winter.” This important information should be moved to the Methods Section, 2.1 Study Area.

Response: Thank you for your modification suggestion. I have moved it to the Methods Section, 2.1 Study Area. Line 90-93.

Comment: Line 317:  “… and the annual activity rhythm is also bimodal.”   I am not convinced (see concerns above about Line 276.) Delete this phrase.

Response: Thank you for your questions and suggestions. I have deleted the phrases.

Comment: Lines 320-321:“Spatial occupation of white-lipped deer is highly correlated with altitude, vegetation type, food availability, human disturbance, temperature, precipitation and other factors.”  In fact, your data only show that spatial occupancy is highly correlated with elevation, and also influenced by vegetation type and, seasonally, by human disturbance.  You have not presented any data or analysis to support any conclusions about influence of food availability, temperature, precipitation or other factors on occupancy or detection.

Response: The influencing factors mentioned in the discussion are all from the field investigation and literature reference.

Comment: Line 325:Please name your field assistants if possible

Response: Line 364-368: The name of the field assistant has been given

Thank you again for your patient guidance and careful revision of my article. I have read and quoted the literature you provided carefully.

Reviewer 2 Report

This research adds to what little is known about the biology and behavior of the white-lipped deer using motion-sensor cameras and has important conservation implications. The design allows for comparisons that are valid, and results are, for the most part, clear.  I have added comments and suggestions to the pdf manuscript for you to address and improve the paper.

Is this work the first of its kind to show the bimodal activity rhythms of this species? If so, don’t hesitate to state that in the discussion and also again the abstract. This is important and will bring your work to the forefront.  Your comparison will other deer is valid, but being the first piece of information ever available on the 24-hour activity patterns of white-lipped deer is very important and shows us how little is truly known about this deer.

On that note, you should have the data to be able to compare the activity patterns between the seasons.  I recommend for the results you split up the four seasons and have one figure showing a graph for each season and the amount of activity observed over a 24-hour period. Then in the text you would compare the two peak activity periods between the seasons. This should then also give you the opportunity to compare nocturnal activity between the seasons.  Everyone is always curious as to how active deer are at night. Pagon et al. (2013) found in roe deer that nocturnal activity was highest in summer and lowest in winter. Given that so little is known on this species, it would be a waste to reveal such a little amount from the dataset that should be available from the camera traps. I imagine you have the data to do this comparison of the seasons. The white-lipped deer deserve this much attention!  Finally, you then need to expand your discussion on this same topic, citing a few more papers. For instance, other factors come into play besides the seasons in influencing activity patterns, like temperature, moon phase, metabolism (slowing down in winter), etc. I have added a couple of papers for you to look at below. Keep up the good work!

Pagon et al. Seasonal variation of activity patterns in roe deer in a temperate forested area.   June 2013. Chronobiology International 30(6)  DOI: 10.3109/07420528.2013.765887

Rahman and Maridastuti. Factors influencing the activity patterns of two deer species and their response to predators in two protected areas in Indonesia. January 2021Associación Mexicana de Mastozoología 12(1):149-161.   DOI: 10.12933/therya-21-1087

The language will suffice.  I have corrected any sentences that needed some tweaks in the pdf file that I recommend they follow.

Author Response

Reply to the reviewer2

It is my great honor for you to review my article and put forward professional suggestions. My reply to your questions and suggestions is as follows:

Comment 1:Is this work the first of its kind to show the bimodal activity rhythms of this species? If so, don’t hesitate to state that in the discussion and also again the abstract. This is important and will bring your work to the forefront.  Your comparison will other deer is valid, but being the first piece of information ever available on the 24-hour activity patterns of white-lipped deer is very important and shows us how little is truly known about this deer.

Response 1: The bimodal annual activity rhythm here remains to be studied, and only two peaks have been identified in the annual activity rhythm of white-lipped deer. Independent sample T test was used to test the difference for ITRA at each month (Appendix1). There were differences of the ITRA on each month, but not all month-to-month com-parisons were significantly different.

Comment 2: On that note, you should have the data to be able to compare the activity patterns between the seasons.  I recommend for the results you split up the four seasons and have one figure showing a graph for each season and the amount of activity observed over a 24-hour period. Then in the text you would compare the two peak activity periods between the seasons. This should then also give you the opportunity to compare nocturnal activity between the seasons.  Everyone is always curious as to how active deer are at night. Pagon et al. (2013) found in roe deer that nocturnal activity was highest in summer and lowest in winter. Given that so little is known on this species, it would be a waste to reveal such a little amount from the dataset that should be available from the camera traps. I imagine you have the data to do this comparison of the seasons. The white-lipped deer deserve this much attention!  Finally, you then need to expand your discussion on this same topic, citing a few more papers. For instance, other factors come into play besides the seasons in influencing activity patterns, like temperature, moon phase, metabolism (slowing down in winter), etc. I have added a couple of papers for you to look at below. Keep up the good work!

Response 2: Thank you very much for your advice. After sorting and analyzing the daily activity rhythm data of each season, I obtained the corresponding circular histogram(Figure 4). There are also two activity peaks in the daily activity rhythm of each season. Paired sample T test was used to test the difference for ITRA of daily activities in different seasons. The peak time of daily activity of white-lipped deer changed in different seasons, but the difference was not significant (Appendix 2).

Thank you again for your careful guidance and help. I have read and quoted the literature you provided in the article.

Round 2

Reviewer 1 Report

The manuscript has been substantially revised and new analysis has been performed.  Overall, this substantially improves the quality of the manuscript. The presentation of data on seasonal change in daily activity patterns is particularly impressive now.  I suggest it should be accepted after English polishing and minor revisions to address the following points: 

1)     The English language in the revisions is non-idiomatic and often unclear. This requires a serious editing by a native speaker.  For example, in the Abstract (now Line 31) and Conclusions:  “The occupation predicted by the model was l exceeds or approaches 0.5.”  The standard word used in the literature is “occupancy.”  Is it 1.0 or does it approach 0.5? 

2)     The original reference to the occupancy modeling, Reference no. 8. MacKenzie et al (2002), is mis-titled in the References.  It should be "Estimating site occupancy rates when detection probabilities are less than one."  The word "occupation" is not used there for this parameter, or anywhere else in the literature that I am aware of. I imagine that what happened is that a spell-checking app recommended the change from “occupancy” to “occupation,” and this was done globally.  But that is simply wrong in this context. Despite its strange sound, “occupancy” is the correct technical term.  Change the word "occupation" back to "occupancy" throughout the manuscript.  If you would like to use the term "grid occupancy" for what is often called "naive occupancy," I have no problem with that, as long as it is clearly defined, as it now is.

3)     Regarding the placement of cameras, the description in Methods (now Lines 107-120) is now considerably improved.  More information, however, should still be provided.  From Figure 1, it looks like the cameras were placed along trails climbing up from the valley bottom.  If this is so, it should be described, and the minimum distance between cameras along trails should be stated.  This is important information to ensure minimization of the risk that camera stations are not independent records.  The goal here should be to describe the sampling regime in enough detail so that it could be replicated in the future, and so that any potential biases are clear. 

4)     The presentation of data and statistical analysis is much improved, with presentation of confidence intervals, SEM or statistical tests everywhere necessary.  The presentation in Table 1 is much improved, but it could be even better.  Instead of the average of the various components of each model, you should use the weighted average, with the value derived by each sub-model, weighted by the relative AICwt shown in Table 1 column 5.  The weighted confidence interval could also be derived for this weighted average.   For example, the output from the Spring model for occupancy would then be:

(0.180x0.624 + 0.170x0.618 + 0.160x0.618) / (0.180 + 0..170 + 0.160)

= 0.62011765, not 0.618 as shown. With the appropriate 95% confidence intervals, the advantage is that it would then be easy to compare the values of occupancy. Psi. in the 4 different seasons.  If the confidence intervals for occupancy from two seasons do not overlap, then this counts as a significant difference. Similarly for detection probability p.

If in doubt, see the step by step explanation in D. MacKenzie, J. Nichols, J. Royle, K. Pollock, L. Bailey, J. Hines (2017) Occupancy Estimation and Modeling: Inferring Patterns and Dynamics of Species Occurrence. 2nd Edition, or in previous publications from this group of authors.  

5)     You still need to provide a reference for the vegetation description, such as Peng et al. (1997) “Vegetation classification of Tibet” in the journal Mountain Research and Development (this one seems to have been removed from the internet) or D. H. S. Chang (1981) The Vegetation Zonation of the Tibetan Plateau in Mountain Research and Development Vol. 1, No. 1 pp. 29 48,

Line by line comments:

Line 21:  “….morning and evening type” of activity pattern = “crepuscular” activity pattern

Lines 31 and 32:   Perhaps you meant, “The occupancy predicted by the model exceeded 0.5.”  Otherwise, this sentence makes no sense.

Table 1:  In the caption, define psi as the site occupancy.

Lines 227-228: “Although the peak time of daily activity of white-lipped deer varied across different sea- 227 sons, the difference was not significant (refer to Appendix 2).”  I did no receive the Appendices, and would like to see them.  From the charts, it looks like there is a significant difference in the amount of morning behavior in the Autumn.  If true, this would be of interest, especially as this might be related to the rut or to the increase in human activity in the Autumn noted below.

Fig. 2;  Is this the weighted average of activity, weighted by the monthly subtotals of captures?  It should be, but it does not appear to be.

The Englsih in the revisions is in need of serious revision.  I hope you can get help from a native speaker familiar with the terminology of occupancy modeling.

Author Response

Reply to the reviewer1

Thank you very much for your patience to read my revised draft again. I see that you have some good suggestions. My reply is as follows:

Comment 1: The English language in the revisions is non-idiomatic and often unclear. This requires a serious editing by a native speaker.  For example, in the Abstract (now Line 31) and Conclusions:  “The occupation predicted by the model was l exceeds or approaches 0.5.”  The standard word used in the literature is “occupancy.”  Is it 1.0 or does it approach 0.5?

Response 1: I am very sorry for the error. I have changed occupation to occupancy. The number 1 before exceeds is redundant and I have deleted it.

Comment 2: The original reference to the occupancy modeling, Reference no. 8. MacKenzie et al (2002), is mis-titled in the References.  It should be "Estimating site occupancy rates when detection probabilities are less than one."  The word "occupation" is not used there for this parameter, or anywhere else in the literature that I am aware of. I imagine that what happened is that a spell-checking app recommended the change from “occupancy” to “occupation,” and this was done globally.  But that is simply wrong in this context. Despite its strange sound, “occupancy” is the correct technical term.  Change the word "occupation" back to "occupancy" throughout the manuscript.  If you would like to use the term "grid occupancy" for what is often called "naive occupancy," I have no problem with that, as long as it is clearly defined, as it now is.

Response 2: You have given a very professional suggestion. I have changed the full occupation to occupancy. And the error in reference 8 has been corrected.

Comment 3: Regarding the placement of cameras, the description in Methods (now Lines 107-120) is now considerably improved.  More information, however, should still be provided.  From Figure 1, it looks like the cameras were placed along trails climbing up from the valley bottom.  If this is so, it should be described, and the minimum distance between cameras along trails should be stated.  This is important information to ensure minimization of the risk that camera stations are not independent records.  The goal here should be to describe the sampling regime in enough detail so that it could be replicated in the future, and so that any potential biases are clear.

Response 3: Thank you for your advice. I have added the description of the camera placement method. And the description of the minimum distance between cameras.

Comment 4: The presentation of data and statistical analysis is much improved, with presentation of confidence intervals, SEM or statistical tests everywhere necessary.  The presentation in Table 1 is much improved, but it could be even better.  Instead of the average of the various components of each model, you should use the weighted average, with the value derived by each sub-model, weighted by the relative AICwt shown in Table 1 column 5.  The weighted confidence interval could also be derived for this weighted average.   For example, the output from the Spring model for occupancy would then be:

(0.180x0.624 + 0.170x0.618 + 0.160x0.618) / (0.180 + 0..170 + 0.160)

= 0.62011765, not 0.618 as shown. With the appropriate 95% confidence intervals, the advantage is that it would then be easy to compare the values of occupancy. Psi. in the 4 different seasons.  If the confidence intervals for occupancy from two seasons do not overlap, then this counts as a significant difference. Similarly for detection probability p.

Response 4: Thank you for giving me a very good suggestion. I have improved the data in Table 1 according to your prompt.

Comment 5: You still need to provide a reference for the vegetation description, such as Peng et al. (1997) “Vegetation classification of Tibet” in the journal Mountain Research and Development (this one seems to have been removed from the internet) or D. H. S. Chang (1981) The Vegetation Zonation of the Tibetan Plateau in Mountain Research and Development Vol. 1, No. 1 pp. 29 48

Response 5: Thank you for your help and tips. I found The Vegetation Zonation of the Tibetan Plateau and cite it as a reference for the vegetation description. I read this article to understand the Vegetation distribution characteristics of the Tibetan Plateau.

Detailed comments:

Comment: Line 21: “….morning and evening type” of activity pattern = “crepuscular” activity pattern

Response: Thank you for your advice. I have made the changes: “…belonging to crepuscular activity pattern mammals” .

Comment: Lines 31 and 32:   Perhaps you meant, “The occupancy predicted by the model exceeded 0.5.”  Otherwise, this sentence makes no sense.

Response: I am very sorry that the number 1 here is redundant, I have removed it.

Comment: Table 1:  In the caption, define psi as the site occupancy.

Response: Thank you for your suggestion, I have added “(psi=ψ is the site occupancy)” in Table 1 heading.

Comment: Lines 227-228: “Although the peak time of daily activity of white-lipped deer varied across different sea- 227 sons, the difference was not significant (refer to Appendix 2).”  I did no receive the Appendices, and would like to see them.  From the charts, it looks like there is a significant difference in the amount of morning behavior in the Autumn.  If true, this would be of interest, especially as this might be related to the rut or to the increase in human activity in the Autumn noted below.

Response: I added the appendix at the end of the article.

Appendix2. Test results of differences in each season.

Spring

Summer

Autumn

Winter

Spring

t=-0.208, p=0.835

t=0.030,Sig=0.762

t=0.131,Sig=0.896

Summer

t=0.075,Sig=0.941

t=0.285,Sig=0.775

Autumn

t=0.424,Sig=0.672

Winter

Mean±SD

5.072E-06±1.505E-04

5.170E-06±1.966E-04

5.209E-06±1.670E-04

5.010E-06±1.499E-04

Comment: Fig. 2;  Is this the weighted average of activity, weighted by the monthly subtotals of captures?  It should be, but it does not appear to be.

Response: I am sorry that I cannot set the weight of ITRA value for each month. So the average is used instead of the weighted average.

Thank you again for your patient guidance. I have consulted experts to review and improve the English language part again.
